# The ALTIUS mission

Didier Fussen, Emmanuel Dekemper, Quentin Errera, Ghislain Franssens, Nina Mateshvili, Didier Pieroux, and Filip Vanhellemont

Royal Belgian Institute for Space Aeronomy, 3, avenue Circulaire, B1180 Brussels, BELGIUM *Correspondence to:* Didier Fussen (Didier.Fussen@aeronomie.be)

#### Abstract.

This article outlines the objectives, concept and expected performance of the ALTIUS (Atmospheric Limb Tracker for the Investigation of the Upcoming Stratosphere) mission in view of the continuation of earth limb measurements for atmospheric science. This type of measurement became rare with the failure of the European ENVISAT mission in 2012 and the number will

further decrease when several Canadian, Swedish and US limb missions will terminate within the next few years. The project is presented in the frame of a small mission initiative based on a micro-satellite platform of the PROBA (Project for On-Board Autonomy) class, with a high agility allowing for atmospheric limb observations in different remote sensing geometries from a low earth orbit. The instrument consists of three independent spectral imagers covering the UV-Vis-NIR ranges.

Recently, the ALTIUS mission has been declared as an element compliant to the ESA Earth Watch programme.

The paper identifies the general scientific context of the project and derives the mission, instrument and scientific products requirements. The general design of the payload and platform systems is discussed. The preliminary data processing chain is presented, from telemetry data to retrieved geophysical profiles, with a complementary data assimilation level. A preliminary assessment of the mission performance is discussed with focus on ozone profile retrievals, which are the main objective of the mission.

# 15 1 Introduction

#### 1.1 Evolution of the Earth's upper atmosphere during the 21st century

It is now accepted that the global and polar depletions of the ozone layer can be attributed to the presence of halogen compounds released by anthropogenic emissions. The Montreal protocol has caused a decrease in the stratospheric halogen load and a slowing of ozone decline is expected to be the natural precursor of a complete ozone recovery towards 2050. There is

presently some experimental evidence that the global mean ozone total column is no longer decreasing with respect to the 1998-2001 period. Also, the ozone stratospheric distribution has been relatively constant during the last decade although both dynamical and chemical processes may contribute to decadal changes in the lower stratosphere. Clearly, the monitoring of ozone stratospheric abundances is of crucial importance in assessing the milestones of a clear recovery process on a global scale (Stocker, 2013).

5

Among important trace gases, methane is very important for its impact on climate through a large radiative forcing effect and the production of stratospheric water vapour. The global increase of about 0.7 ppm in 1800 AD to 1.8 ppm nowadays is difficult to interpret because of the diversity of the sources: wetlands, enteric fermentation, fires and rice agriculture. The odd hydrogen family, HOx, contains all active species, i.e. radicals that are involved in catalytic cycles that destroy Ox. The HOx radicals are derived primarily from the oxidation of water vapour in the stratosphere and it is therefore essential to understand and to menitor the interpret primarily from the oxidation of water vapour in the stratosphere and it is therefore essential to understand

and to monitor the intrusion of water vapour into the stratosphere, especially in the region of the tropical tropopause (Hegglin, 2013). Similarly, the NOx family is known to play an essential catalytic role in ozone destruction with a strong diurnal cycle that requires day- and night-time measurements for a complete understanding.

Much remains to be learned about polar stratospheric clouds (PSC) and we do not know enough about particle sizes, crystal morphology and even composition. The same uncertainty holds for polar mesospheric clouds (PMC), similar in appearance to thin cirrus clouds, but located at a much higher altitude, from 80 to 87 km near the mesopause. PMCs only occur at high latitudes during summer (a few weeks before and after the solstice), when the mesosphere becomes extremely cold (with temperatures as low as 100 Kelvin) (Deland, 2007). Their increasing occurrence may be a sign of climate change in the upper atmosphere.

On the other side, the scientific community is requiring an urgent response to the present lack of atmospheric remote sounders (see Figure 1) with a high vertical resolution. Since the end of ENVISAT in 2012, some aging limb missions/instruments are still operating (ODIN/OSIRIS (Llewellyn, 2004), AURA/MLS (Froidevaux, 2008), ACE/FTS-MAESTRO (Dupuy, 2009), OMPS-NPP/LS (Moy, 2016)) but they are all beyond their nominal lifetime. In the near future, ISS/SAGE-III (2016) and JPSS-2/OMPS-LP (2022) are the only approved missions so far and there is no guarantee concerning a possible overlap between them.

them.

This article presents the concepts of a mission called ALTIUS (Atmospheric Limb Tracking for the Investigation of the Upcoming Stratosphere) that was successfully proposed by the Belgian Institute for Space Aeronomy (BISA) to the Belgian Scientific Policy Office (BELSPO) and that has been recently approved by the European Space Agency (ESA) as an element of the ESA Earth Watch programme. The general description reads as "ALTIUS aims at the development, launch, in-orbit

- operation, data processing, data archiving and products distribution of a limb sounder mission based on a small satellite. Its main objective is the monitoring of the 3-D distribution and evolution of stratospheric ozone at high vertical resolution. The versatile instrument, which will image the Earth's limb in the near UV, Visible and Near Infrared spectral regions, in combination with the agility provided by the spacecraft platform can allow measurements of concentration profiles of other species as well as of aerosol extinction vertical profiles."
- The present paper gives an overview of the objectives, concept and expected performances of the ALTIUS mission as proposed by BISA, on its way to a preliminary design review (PDR), expected at the end of 2017.

## 1.2 Scientific objectives of the ALTIUS mission

In view of the need for atmospheric composition measurements on a global scale and with a high vertical resolution, it is useful to summarize the mission objectives in a series of scientific requirements (SR) and to group them into subsets respectively identified as of "mandatory", "important" and "relevant" weight.

#### 5 1.2.1 Mandatory scientific requirements

- SR1 Global and long-term vertically resolved ozone data sets of observations in the stratosphere are urgently required to assess model behaviour and test model predictions, particularly in the upper troposphere lower stratosphere (UTLS) domain, in polar regions and in the Southern Hemisphere. Today, the ozone recovery, which is probable but not yet confirmed, is still an open question.
- 10 SR2 The measurement of ozone profiles in the middle stratosphere at 5 % accuracy level has to be continued to reinforce the significance of the existing climatologies. A special effort is needed in the UTLS region to improve the determination of trends and to detect increase in upwelling, a predicted climate change effect.
  - SR3 The spread of existing ozone profile data in ozone hole conditions is not satisfactory. Limb observations should allow determining spatial concentration gradients across and along the vortex while detecting PSC's for data screening and correlative analyses, during all seasons (including polar night).
  - SR4 ALTIUS should also provide highly resolved vertical profiles of mesospheric  $O_3$ , a key species to understand the atmospheric coupling, not yet well understood, between the lower atmosphere and the upper stratosphere, mesosphere and lower thermosphere.

#### 1.2.2 Important scientific requirements

- SR5 ALTIUS will provide highly resolved vertical profiles of  $NO_2$  from the UTLS to the upper stratosphere, mesosphere and lower thermosphere at different local times. It will provide intercomparison of  $NO_2$  and  $O_3$  abundances. It will also focus on the dynamics and the chemistry of strong  $NO_2$  enhancements in the upper stratosphere-mesosphere.
  - SR6 The mission will provide global vertical profiles of  $H_2O$  and  $CH_4$ , with a special focus, for water vapor, on the tropical tape recorder and vortex dehydration. A major objective for these interrelated species will be the measurement of their trends in the lower stratosphere.
- 25

15

- SR7 ALTIUS will provide extinction profiles and particle size distributions of stratospheric aerosol by observing vertical extinction profiles in the UV-Vis-NIR wavelength ranges. Depending on the volcanic state of the atmosphere, it will assess the relaxation times of the volcanic contribution or the evolution of the background non-volcanic contribution.
- SR8 It is important to measure the frequency of PSC occurrences inside the polar vortex and their temperature dependence.

#### **1.2.3** Relevant scientific requirements

- SR9 ALTIUS will measure the trend and the phase in the occurrence of PMC's as well as their median altitude and their horizontal extent, in both hemispheres, around the summer solstice.
- SR10 ALTIUS will measure OCIO, BrO and NO<sub>3</sub>, that are important minor trace gases involved in the stratospheric chemistry.
- SR11 In solar occultation mode, by making use of imaging techniques at large S/N ratio, ALTIUS will retrieve density and temperature profiles up to the mesosphere, from refraction angle measurements.
  - SR12 Horizontal concentration gradients of relevant trace gases will be observed in "along track" and "across track" geometries, allowing for tomographic retrievals during successive revolutions.

## 1.3 ALTIUS scientific product and mission requirements

- In Table 1, we report the target scientific product requirements related to the abovementioned scientific objectives. The baseline assumption for ALTIUS supposes a limb sounder onboard a micro-satellite platform (a PROBA class carrier was pre-selected for its agility and pointing performance). This choice ensures a global coverage (including polar regions) if launched in a heliosynchronous low earth orbit (LEO), and the capability to achieve the highest vertical resolution. As a threshold requirement for the use of ALTIUS data in present assimilation models, the global coverage should be sampled on a grid finer than 5-10
- degrees in latitude and 10 degrees in longitude.

Global coverage and high vertical resolution can be achieved by combining several observation modes based on the interaction of the light emitted by a celestial body and the atmosphere, such as the limb-scattered solar light, solar (and lunar) occultations, stellar (and planetary) occultations. Indeed, bright limb measurements offer a large sampling on the day side of the orbit, while occultations add a number of measurement points on the night side.

- This multimode capacity requires some agility, autonomy and stability from the satellite, which is precisely what has been demonstrated by the PROBA platform. A further constraint is that ALTIUS must be the only payload with pointing requirements onboard. Limb measurements (not based on atmospheric emissions) are very sensitive to the tangent altitude registration of the line-of-sight(LOS), i.e. the closest point to the local geoid, especially where the concentration profiles show large vertical gradients. Previous limb-scatter instruments have experienced serious tangent height misregistration issues (SCIAMACHY,
- OSIRIS, OMPS). Occultations also need to keep the light source in the field-of-view (FOV). The original approach proposed for ALTIUS is to use an imaging system with a FOV matching the apparent size of the bright limb (0-100km). With this method, in-flight calibration methods are more easily implemented to solve the pointing issue. The entire atmosphere is probed at once, which is an advantage compared to scanning systems which take tens of seconds to complete the scan (and lose in along-track resolution). In addition, inertial pointing to the occulted celestial bodies is done without the need for complex light
- source tracking systems (as it was the case for GOMOS on ENVISAT).

#### 2 Instrument requirements and operation concepts

#### 2.1 Instrument concept and requirements

In order to solve the tangent height misregistration problem, ALTIUS is designed as a limb imager for which the field-of-view can be calibrated by different techniques. Assuming a rectangular FOV, the atmospheric limb shall be imaged between 0 and

- 5 100 km (about 34 x 34 mrad) although stellar and planetary occultations can be performed with a smaller FOV of 3.4 x 3.4 mrad. The pixel FOV will be less than or equal to 0.2 mrad allowing for a vertical sampling better than 0.6 km. The overall instrument Modulation Transfer Function (MTF) at 2.5 cycles/mrad will be greater than 20 % and, for stellar occultations, more than 20 % of the energy will be concentrated in a single pixel.
- Clearly, limb imaging imposes stringent pointing requirements on the system. If we consider a Cartesian Boresight Reference
  Frame (BRF) centered at the detector with the Z axis pointing to the limb horizon, the X and Y axis respectively perpendicular and parallel to the horizon, we can define angular uncertainties related to the rotations R<sub>Z</sub>, R<sub>X</sub> and R<sub>Y</sub> around these axes. The corresponding requirements are reported in Table 2 for the Mean Performance Error (MPE), Mean Knowledge Error (MKE), Relative Performance Error (RPE), Relative Knowledge Error (RKE), Performance Drift Error (PDE) and Knowledge Drift Error (KDE) (see Ott (2011) for an exact definition of these error terms).
- ALTIUS may be described as an imaging spectrometer of moderate spectral resolution. The instrument will be a tuneable spectral imager capable of observing the atmospheric limb in the UV (250-400 nm), VIS (420-800 nm) and NIR (800-1800 nm) domains, with a spectral width (depending quadratically on wavelength) always better than 10 nanometers for VIS and NIR and better than 2.5 nm for UV. A distinct spectral tuneable element will filter observations in each of these three wavelength ranges.
- In general, the instrument will operate at different measurement rates corresponding to the different observation geometries. For limb scattering observations, the along track sampling rate is consistent with the effective optical length along the line-ofsight (about 500 km). It is important to notice that a limb pointing instrument is unable to achieve a very large swath like a nadir looking sensor due to the Earth curvature. In Table 2, we report the typical number of images (spectral snapshots) and observations (minimum ensemble of images at different wavelengths to retrieve the spectral absorber profiles).
- In Table 3, we identify the useful spectral windows to measure the concentration profiles of the main target gases together with the required maximum spectral width, minimum SNR and maximum acceptable pointing error. In limb mode, a typical set of 10 wavelengths per channel will be recorded nominally in ten seconds to obtain sufficient spectral information content with a maximal geographical resolution. However, it will be possible to increase the measurement time up to 50 seconds (or even 100 seconds in the worst case) to improve the S/N ratio. Pixel binning (up to the full detector row) will also be made
- 30 possible for the same purpose at the price of a reduced horizontal resolution.

# 2.2 Instrument design

If the goal was to take spectral pictures at fixed wavelengths, a filter wheel could have been used. But then the wavelengths would be frozen, with a supplementary risk of mechanical failure in space environment. To overcome these issues, ALTIUS will use a tunable approach based on:

- Acousto-optical tunable filters (AOTF) for the VIS and IR channels.
  - Fabry-Pérot interferometers (FPI) for the UV channel.

AOTFs are small birefringent crystals (typically a few cubic centimetres) serving as interaction medium between the incoming light and an acoustic wave propagating in the crystal (Dekemper, 2012; Chang, 1974; Xu, 1992). By carefully selecting the wave frequency, the acousto-optic interaction turns induces a Bragg diffraction regime that deflects light with a specific

- wavelength away from its incident direction (by a few degrees). The diverted beam can then be collected by an off-axis optical detector. Used inside an imaging system, such a device offers many advantages: it is small and lightweight, contains no moving parts, it consumes only 1 to 3 watts and can be tuned to the required wavelength in a few milliseconds. An AOTF works over a broad spectral range (hundreds of nanometres) with a variable bandwidth (quadratic in wavelength) that can be optimized by design (from sub-nm to 10 nm).
- As AOTF crystals in the UV region are not yet mature enough to be considered as a suitable technology for space application, the filter element for that region will consist of a cascade of Fabry-Pérot interferometers.

ALTIUS benefits from a relatively simple optical design. The three spectral channels share the same idea: an aperture collects the (ir)radiance of the scene which is then reflected by a number of mirrors to form an image onto a detector array. Located approximately half-way between the aperture and the detector, a tuneable spectral filter captures a fraction of the light spectrum

while the rest of the incident beam is blocked by the combined action of a beam stop and cross-oriented polarizers (AOTF), or by the filter itself (Fabry-Pérot).

Each channel is made of reflective optics and contains the same set of functional modules:

- A mechanical structure
- A front-end optics group (FEO), to guide the incoming light towards the spectral element
- A spectral filter (AOTF module or FPI assembly)
  - A back-end optics group (BEO) to focus the filtered image on the detector
  - A detector with optional cooling capacity
  - A driving and read-out electronic board

At the detector level, a 512x512 CMOS image sensor is proposed, suitable for atmospheric absorption measurements as a 30 hyper-spectral application in the UV and visible wavelength ranges. It has a programmable or automatic high dynamic range:

two full well factors are foreseen with a ratio of 15 and a linearly or exponentially varying row integration time along the atmospheric radiance gradient, capable of at least 10 frames per second, and a wide operational temperature domain. For the NIR detector, the required SNR imposes a 190 K working temperature which will be obtained by using a Stirling cooler. Such a device is capable of significantly reducing the chip operating temperature at the expense of a power consumption of about 5 W and a limited lifetime that will have to be taken into account in the mission scenario.

An extended set of on-ground calibrations is foreseen to fully characterize the sensor. Instrument performance will be checked at an accuracy level compatible with the sensor requirements (typically 1/10 pixel for LOS and pointing, 1-3 % for radiometric quantities, 1/10 of spectral bandwidth for wavelength). In particular, the following characteristics will be calibrated:

- LOS calibration per channel.
- LOS and pixel mapping for FOV, IFOV and distortion.
  - Temperature dependence of pixel mapping.
  - MTF and point spread function.
  - Radiometric response: PRNU, SNR, pixel non-linearity and dynamic range.
  - Straylight.
- Spectral response: wavelength calibration and temperature dependence, bandwidth and out-of-band rejection.

So far, the ALTIUS sensor design has been performed through many technical studies to demonstrate its feasibility and its compliance to the payload requirements. All these study phases have been thoroughly reviewed by several technical committees. No showstoppers were found whereas missing studies or optimizations have been identified: a detailed description of the payload/platform interface, a global optimization of the UV channel, the detector layout, the payload electrical design, a consolidated thermal analysis focused on the assessment of the thermo-elastic effects (and their impact on the pointing requirements) and a comprehensive straylight analysis. All these missing technical studies will have to be conducted before the Preliminary Design Review (PDR) presently foreseen at the end of 2017 and will support several publications about a detailed and consolidated description of the sensor.

# 2.3 Platform, orbit and operations

# 25 2.3.1 Platform

The payload has been designed to fit in an evolved PROBA platform design, as shown in Fig. 2 (internal view) and Fig. 3 (external view) below. The platform generic design has been modified in order to implement a propulsion system, to provide the necessary accommodation volume for the instrument as well as sufficient power to sustain the mission, and to remain within volume constraints for a set of realistic launchers compatible with the ALTIUS mission.

# 2.3.2 Orbit

Heliosynchronicity is a standard requirement for a limb-scattering instrument as air masses are sounded at a constant local solar time, while it makes the radiative transfer problem of the limb-scattered light more easily addressed by narrowing the range of solar angles to a small subset (drift of the orbital local ascending node will be limited to 0.5 hours over the mission lifetime).

A revisit time of about 3 days was selected (similar to ENVISAT) with some loose requirements on the exact overlapping of the S/C ground tracks. For a low earth circular orbit (LEO) at an altitude of 680 km, this allows for an interesting trade-off between a short revisit time and a dense geographical sampling grid (about 800 km between successive tracks at equator) that matches the typical horizontal resolution of data assimilation models.

# **2.3.3** Baseline operations

- During 90 % of the mission time, ALTIUS will be operated according to a so-called "baseline" scenario depicted in Fig. 4:
  - 1. On the dayside (bright limb), it observes scattered solar radiation in the backward direction with respect to the velocity vector
  - 2. When approaching the terminator, a spacecraft manoeuvre brings the Sun into the FOV to observe a sunset occultation in inertial pointing mode
- 3. On the nightside (dark limb), 5 to 10 bodies (stars and planets) are selected and rallied to observe their occultation
  - 4. At the terminator again, ALTIUS observes a sunrise occultation
  - 5. Back to step 1, for a new cycle of observations.

All system resources (energy, thermal budget, telemetry,..) have been optimized from this scenario. However, 10 % of the mission scenario is spare for less regular activities, including instrument calibration campaigns. The sequence of night objects

- (stars, planets and the Moon) selected for candidate observations was optimized with respect to the number of necessary satellite manoeuvres. In Fig. 5, we report the ensemble of successive observations for one satellite revolution in the baseline scenario, with successive sunrise occultation, limb scattering observations in dayside, sunset and, in the nightside, stellar, planetary and lunar occultations. A full day of ALTIUS observations is plotted in Fig. 6, where the geolocations of observations are progressing westward by about 24 degrees between two successive revolutions. The simulation was performed for a quasi-
- circular (eccentricity=  $1.05 \, 10^{-3}$  degree) orbit with a semi-major axis of 7050 km and an inclination of 98.07° degrees. Fig. 7 and 8 respectively represent the seasonal evolution of bright limb and solar occultation geolocations, the latter ones being confined to rather high latitudes as a consequence of the almost polar orbit inclination. On the other hand, lunar occultations in dark limb are also possible and allow for tropical observations (see Fig. 9). Finally, we also report the geolocations of brightest star and planet occultations in Figs. 10 and 11.

#### 2.3.4 Special observation modes

As a serendipitous mode of observation, it shall also be possible to perform tomography of a species 3D distribution field in the low latitude regions. Due to the remarkable (and fortuitous) fact that the azimuthal angular distance to the side horizon is close to the angle by which the Earth will have rotated at the next LEO revolution, it is possible to combine a backward limb observation followed by a dedicated sideward limb observation at the consecutive orbit. This mode is called tomographic because the same location is observed with a small delay of about 1.5 hours, from two almost orthogonal directions in equatorial

regions, which allows for a 3-D inversion of the geophysical fields.

Even if the mission scenario of ALTIUS is driven by the baseline observation modes described above, the platform high manoeuvrability enables a pointing in many directions for dedicated and limited campaigns. Taking into account the total

10

5

available power, thermal budgets, data transmission, exposition of star trackers to direct solar light, etc, the feasibility and programming of these campaigns will require case-by-case studies. Therefore they will only be implemented after successful commissioning. As a non-exhaustive list of dedicated observation modes, we can mention:

- Specific airglow emissions
- Lunar occultations
- 15 - Inter-validation with nadir looking instruments
  - Volcanic events
  - "Colour of clouds" measurements
  - Effect of solar proton events
  - "Tangent" occultations (celestial body doesn't set for a very obique occultation and is grazing the horizon)
- 20
- Polarization-sensitive measurements (by rotating the satellite)
  - Photochemical effect of solar eclipses

#### Multimode observations 2.4

#### 2.4.1 Bright limb images

In Fig. 12, bright limb radiance simulations have been obtained from the radiative transfer code MODTRAN 5, for a US 25 standard atmosphere 1976 (mid-latitude summer model) with a background stratospheric aerosol load and no clouds. The tangent point is located at 45°N, 0°E. ALTIUS looks at this location from a sun-synchronous orbit at 10:30 local time at descending node (LTDN), backward with respect to the platform velocity vector. The radiance was calculated for 19 tangent altitudes and exhibits a large dynamical range with respect to the tangent altitude. Subsequently, raw ALTIUS images were simulated for any wavelength in each channel. When possible, the pixel gain and the exposure time have been optimized to