# Peer review of "The ALTIUS mission"

_Atmospheric Measurement Techniques, 2016_

## Referee Comment (RC1) · Anonymous Referee #1 · 25 Aug 2016

In this paper the authors have laid out their very ambitious plans for ALTIUS, the new ESA Earth Watch mission. It is important that the type of information presented within this document reach a wide audience and publication in a journal like AMT may be acceptable. I will leave that to the editor to make the final decision.

I am happy to accept this work with minor corrections but I do want to make it very clear that this is not a scientific paper in the typical sense. It provides a very brief description of all aspects of the ALTIUS mission including the platform, orbit, instrument, scientific goal, methods and algorithms to produce data products, etc. These are very interesting pieces of information and should be made available to as wide an audience as possible. However, in almost every case there is not enough information presented to allow a reader to understand the difference between a plan on a wish list and realizable fact. I do not blame the authors for this as that level of detail would increase the length of the paper orders of magnitude beyond a manageable size. I assume that some of

the information exists within a pile of ESA documentation that extends many metres in height. I also assume that some of the ideas presented within this paper have not yet be substantiated and remain just ideas.

This paper outlines the author's wish list should ALTIUS work up to expectation. I very much hope it does, as it will add valuable information for those of us who study the stratosphere. However, as it lacks sufficient detail, this paper gives me no indication that ALTIUS will work. It simply states what ALTIUS produce if it works perfectly. I actually have some serious doubts about many of the statements made within the paper but if the goal of the paper is to describe the instrument/mission and state a wish list I don't believe it is my job to highlight these doubts. If the "for your information" format of the paper is acceptable to the editor then it is acceptable to me.

I have noted some minor issues that should be addressed.

1) In the orbit section it would be used to know the anticipated LTAN.

2) line 16 on page 10 contains too many "smalls".

3) The quality of Figures 5, 6, 8, 9, 10, 11 are 19 is not acceptable. They neither print well nor show up well on a normal size monitor.

4) I do not understand the last sentence in the figure caption of Figure 5.

5) In the caption for Figure 7 it says the maximal solar zenith angle is about 70 degrees. Why is this?

6) What is plotted in Figure 7?

7) Figure 10 needs a colour scale or a description of what the colours mean.

8) The discussion around Figure 15 should involve a little more information on the expected noise levels for solar occultation measurements. The noise on the limb radiance measurements in Figure 13 leads me to suspect that the noise in the solar occultation measurements will also be very significant.

9) The discussion around Figure 20 is not sufficient to understand the information content of the figure.

I wish the authors the best of luck with their very important ALTIUS mission!

―――――――――――――――

---

## Author Comment (AC1) · 28 Sep 2016

[General]REF 1 I am happy to accept this work with minor corrections but I do want to make it very clear that this is not a scientific paper in the typical sense. It provides a very brief description of all aspects of the ALTIUS mission including the platform, orbit, instrument, scientific goal, methods and algorithms to produce data products, etc REPLY: We agree that the paper does not provide data or even the performance level of a final calibrated instrument. As correctly mentioned by the referee, it out-of-scope for a single generic paper and, for some technical studies, not mature enough. For instance, the straylight budget is not yet known and can only be performed once the final instrument design has been completed. However, science not only consists of validated data but also of new ideas (and not simple wishes) capable of tackling the need of atmospheric limb data. The main ideas proposed for the ALTIUS mission are: 1) the concept of spectral imaging in limb scattering observations that should allow for solv-

---

## Referee Comment (RC2) · Anonymous Referee #2 · 10 Oct 2016

As far as I can ascertain, version 3 of the manuscript does not differ significantly from the one (version 1) that I previously reviewed. I stand by those review comments - there is no need to repeat them.

Summary: The submitted manuscript is clearly an overview, emphasizing breadth over detail. And given the recently upgraded status of ALTIUS, such an overview should be published. However, this level of description is usually published in proceedings, e.g. IEEE, SPIE, or other conference proceedings. I note the authors have not referenced any prior publications related to ALTIUS. If this is because none exist, I suggest the authors use proceedings to provide this mission overview. If a prior publication(s) does exist, I recommend they authors wait until more technical detail can be provided before re-submission to AMT.